# Quantifying the nonclassicality of pure dephasing

Hong-Bin Chen [1,2,3], Ping-Yuan Lo [4], Clemens Gneiting[5], Joonwoo Bae [6], Yueh-Nan Chen [1,2] & Franco Nori [5,7]

One of the central problems in quantum theory is to characterize, detect, and quantify quantumness in terms of classical strategies. Dephasing processes, caused by non-dissipative information exchange between quantum systems and environments, provides a natural platform for this purpose, as they control the quantum-to-classical transition. Recently, it has been shown that dephasing dynamics itself can exhibit (non)classical traits, depending on the nature of the system-environment correlations and the related (im)possibility to simulate these dynamics with Hamiltonian ensembles–the classical strategy. Here we establish the framework of detecting and quantifying the nonclassicality for pure dephasing dynamics. The uniqueness of the canonical representation of Hamiltonian ensembles is shown, and a constructive method to determine the latter is presented. We illustrate our method for qubit, qutrit, and qubit-pair pure dephasing and describe how to implement our approach with quantum process tomography experiments. Our work is readily applicable to present-day quantum experiments.

[1] Department of Physics, National Cheng Kung University, Tainan 70101, Taiwan. [2] Center for Quantum Frontiers of Research & Technology, NCKU, Tainan 70101, Taiwan. [3] Department of Engineering Science, National Cheng Kung University, Tainan 70101, Taiwan. [4] Department of Electrophysics, National Chiao Tung University, Hsinchu 30010, Taiwan. [5] Theoretical Quantum Physics Laboratory, RIKEN Cluster for Pioneering Research, Wako-shi, Saitama 351-0198, Japan. [6] School of Electrical Engineering, Korea Advanced Institute of Science and Technology (KAIST), 291 Daehak-ro, Yuseong-gu, Daejeon 34141, Republic of Korea. [7] Physics Department, University of Michigan, Ann Arbor, MI 48109-1040, USA. Correspondence and requests for materials should be addressed to H.-B.C. (email: hongbinchen@phys.ncku.edu.tw) or to Y.-N.C. (email: yuehnan@mail.ncku.edu.tw)

The boundary between the quantum and the classical world has always been a fundamental issue in quantum mechanics[1–4]. An operationally viable way to demonstrate the genuine quantum nature of an experiment relies on the impossibility to mimic certain statistical properties of interest by using a "classical strategy". According to this logic, the quantum nature of an experiment is only convincingly demonstrated if the experimental statistics cannot be mimicked by the classical strategy; thus excluding any loophole to explain the statistics with a classical model.

For example, under the assumptions of realism and locality, Bell[5] derived an inequality for correlations between the statistics of measurements on a bipartite system. Whenever the inequality is violated, one cannot reproduce the correlations by using a local hidden variable model, the latter serving as the classical strategy for mimicking the measurement statistics. Another important paradigm is the quantumness of a boson field, which is formulated in terms of the Wigner function or the Glauber–Sudarshan $P$ representation[6–8]. Whenever these functions exhibit negative values, the classical explanation in terms of a probability distribution over phase space fails to represent the boson field.

Following this spirit, one may ask for a classical strategy to frame the "quantumness" of open system dynamics. This question has been addressed in different ways. In these approaches, specific properties of system states, e.g., Wigner functions with negativities, violation of Leggett–Garg inequality, non-stochasticity of dynamical processes, or detection of quantum coherence, are identified as indicators of nonclassicality and monitored during the temporal evolution[9–16].

Alternatively, we propose to take the presence or absence of quantum correlations between system and environment as a signature for the quantum nature of the open system dynamics. As was shown recently[17], such presence or absence of non-classical system–environment correlations is intimately linked to the (im)possibility to simulate the open system dynamics with a Hamiltonian ensemble (HE), which may thus serve as the classical strategy to witness the nonclassicality of the open system dynamics. HEs, which are also used to describe disordered quantum systems, attribute to each member of a collection of (time-independent) Hamiltonians a probability of occurrence, giving rise to an effective average dynamics.

Finding a simulating HE certifies that the open dynamics is classical. The nonexistence of a simulating HE, on the other hand, can be proven by the necessity to resort to a HE accompanied by negative quasi-distributions. Although being conceptually clear, as was shown in ref. [17] for the example of an extended spin-boson model, this is technically highly nontrival in general; especially for high dimensions. For example, the closely related problem of random-unitary decomposition can in general merely be numerically implemented[18]. An efficient approach appears desirable.

On the other hand, analyzing dephasing is essential for the improvements of quantum information science and quantum technologies. Besides its fundamental relevance for the quantum-to-classical transition[19–21], classicality of the dynamics, reflected by the existence of a simulating HE, can then be related to the in-principle possibility to correct errors caused by the HE[22]. Furthermore, it also constitutes one of the main obstacles in the fabrication and manipulation of quantum information devices[23–28]. Different implementations for the simulation of controlled pure dephasing[29–31] and its mitigation[32–36] exist. Other experiments highlight the potential of decoherence or pure dephasing to contribute positively to certain quantum information tasks, such as entanglement stabilization[37] or entanglement swap[38].

Here, we introduce a measure of nonclassicality for pure dephasing dynamics, i.e., we focus on situations where dephasing constitutes the sole dynamical agent. We begin with recasting any HE into a canonical form; within this framework, each HE is composed of the same canonical set of Hamiltonians, such that the accompanying (quasi-)distribution fully characterizes the HE. Let us remark that one can interpret the resulting representation as a random rotation model, since it is a (quasi-)distribution of rotations induced by the Hamiltonians. We also prove its existence and uniqueness. This promotes it to a faithful representation of the pure dephasing dynamics and allows us to unambiguously quantify the nonclassicality. Additionally, we outline a systematic procedure to retrieve (quasi-)distributions for pure dephasing and elaborate our ideas for qubit, qutrit, and qubit-pair examples. Finally, we also discuss the implementation of our approach with the quantum process tomography experiments to show the ready applicability to present-day quantum experiments.

## Results

**Averaged dynamics of Hamiltonian ensembles.** A HE $\{(p_\lambda, \widehat{H}_\lambda)\}_\lambda$ is a collection of Hermitian operators $\widehat{H}_\lambda$ acting on the same system[17,39], where each member Hamiltonian is drawn according to the probability distribution $p_\lambda \geq 0$. A system $\rho_0$, isolated from any environment, is sent into a unitarily-evolving channel $\rho_\lambda(t) = \widehat{U}_\lambda \rho_0 \widehat{U}_\lambda^\dagger$, with $\widehat{U}_\lambda = exp\left[-i\widehat{H}_\lambda t/\hbar\right]$ for a chosen $\widehat{H}_\lambda$ according to $p_\lambda$. Then, the dynamics of the averaged state $\bar{\rho}(t)$ is given by the unital map

$$\bar{\rho}(t) = \mathcal{E}_t\{\rho_0\} = \int p_\lambda \widehat{U}_\lambda \rho_0 \widehat{U}_\lambda^\dagger d\lambda. \qquad (1)$$

Even though each single realization $\rho_\lambda(t)$ evolves unitarily, the averaged state $\bar{\rho}(t)$ exhibits incoherent behavior[39–42]. A seminal and intriguing example is a single qubit subject to spectral disorder with HE given by $\{(p(\omega), \hbar\omega\hat{\sigma}_z/2)\}_\omega$, then the averaged dynamics describes pure dephasing:

$$\bar{\rho}(t) = \begin{bmatrix} \rho_{\uparrow\uparrow} & \rho_{\uparrow\downarrow}\phi(t) \\ \rho_{\downarrow\uparrow}\phi^*(t) & \rho_{\downarrow\downarrow} \end{bmatrix}, \qquad (2)$$

with the dephasing factor $\phi(t) = \int p(\omega)e^{-i\omega t}d\omega$ being the Fourier transform of the probability distribution $p(\omega)$.

The pure dephasing in Eq. (2) is a consequence of the commuting member Hamiltonian $\hbar\omega\hat{\sigma}_z/2$ in the ensemble. Each Hamiltonian induces a unitary rotation about the $z$-axis of the Bloch sphere at angular velocity $\omega$. This gives rise to an intuitive interpretation of pure dephasing in terms of random phases: each component rotates at different angular velocity $\omega$ and hence possesses its own time-evolving phase. Consequently, the phase of the averaged system gradually blurs out.

Note that $p(\omega)$ is the probability distribution of the angular velocity and qualitatively characterizes the "randomness" of the random rotation. Whenever $p(\omega)$ is specified, the dynamics is uniquely determined via the Fourier transform in Eq. (2). This is also in line with our classification of such pure dephasing as classical[17] since it is a statistical mixture of rotations at different angular velocities. Meanwhile, the experimental simulation of pure dephasing is implemented in a similar spirit[29–31].

**Canonical Hamiltonian-ensemble representation.** Although $p(\omega)$ is particularly representative for characterizing qubit pure dephasing, it is obvious that, in general cases with non-commuting or higher dimensional member Hamiltonians $\widehat{H}_\lambda$, the Fourier transform in Eq. (2) is not applicable. We are, therefore, spurred to explore the canonical Hamiltonian-ensemble representation (CHER) as a generalized representation of an averaged dynamics.

To fully understand the CHER, we first observe that, since both $\widehat{H}_\lambda$ and density matrices $\rho$ are Hermitian, they are elements in the Lie algebra $\mathfrak{u}(n) = \mathfrak{u}(1) \oplus \mathfrak{su}(n)$, which are spanned by the identity $\{\hat{I}\}$ and $\{\widehat{L}_m\}_m$ of $n^2 - 1$ traceless Hermitian generators, respectively. Then $\widehat{H}_\lambda \in \mathfrak{u}(n)$ is a linear combination $\widehat{H}_\lambda = \lambda_0 \hat{I} + \sum_{m=1}^{n^2-1} \lambda_m \widehat{L}_m = \lambda_0 \hat{I} + \boldsymbol{\lambda} \cdot \widehat{\mathbf{L}}$, where $\lambda_0 \in \mathbb{R}$ and $\boldsymbol{\lambda} = \{\lambda_m\}_m \in \mathbb{R}^{n^2-1}$.

Since the dynamics is a linear map acting on $\rho$, invoking to the adjoint representation (see Methods and Supplementary Note 1), we can assign each generator $\widehat{L}_m$ a linear map $\widehat{L}_m \mapsto \widetilde{L}_m \in \mathfrak{gl}(\mathfrak{u}(n))$, with its action $\widetilde{L}_m(\bullet) = [\widehat{L}_m, \bullet]$ defined in terms of the commutator.

With the above mathematical setup, given a HE $\{(p_\lambda, \widehat{H}_\lambda)\}_\lambda$, one can consider the probability distribution $p_\lambda$ as a CHER of an averaged dynamics $\mathcal{E}_t$, in the sense that Eq. (1) can always be recast into a Fourier transform from $p_\lambda$, on a locally compact group $\mathcal{G}$ characterized by the parameter space $\lambda = \{\lambda_0, \boldsymbol{\lambda}\}$, to the dynamical linear map $\mathcal{E}_t^{(\widetilde{L})}$:

$$\mathcal{E}_t^{(\widetilde{L})} = \int_{\mathcal{G}} p_\lambda e^{-i\widetilde{\lambda L}t} d\lambda. \tag{3}$$

Note that we have set $\hbar = 1$ for symbolic abbreviation. Similarly, we can also express $\rho = n^{-1}\hat{I} + \boldsymbol{\rho} \cdot \widehat{\mathbf{L}}$ in terms of a column vector $\rho = \{n^{-1}, \boldsymbol{\rho}\}$, the action of $\mathcal{E}_t$ on $\rho$ is then the usual matrix multiplication $\mathcal{E}_t\{\rho\} = \mathcal{E}_t^{(\widetilde{L})} \cdot \rho$ [see Supplementary Note 2 for the proof of Eq. (3)].

We emphasize that, compared with Eq. (1), the Fourier transform formalism (3) is a powerful tool in the following proof of uniqueness and establishment of our procedure. It also highlights our exclusive focus on the dynamics alone, regardless of the system state. Additionally, it provides further insights into the nature of CHER and the connection to the process nonclassicality, in terms of a random rotation model. In such interpretation, different components rotate about different axes, defined by the generators $\left\{\widehat{L}_m\right\}_m$. Moreover, $p_\lambda$ is the distribution function of the random rotations over the $n^2$-dimensional Euclidean space. This interpretation is consistent with the random phase model in the case of qubit pure dephasing (2).

**HE simulation and process nonclassicality**. So far we have discussed the averaged dynamics of an isolated system, in the absence of any environment, governed by a HE. Conversely, to discuss the nonclassicality of an open system dynamics reduced from a system–environment arrangement, we should construct a simulating $\{(\wp_\lambda, \widehat{H}_\lambda)\}_\lambda$ for a given unital dynamics.

An autonomous system–environment arrangement is characterized by a time-independent total Hamiltonian $\widehat{H}_T$ and evolves unitarily with $\widehat{U}_T = \exp[-i\widehat{H}_T t]$. We have shown that[17], if the total system $\rho_T(t) = \widehat{U}_T \rho_T(0) \widehat{U}_T^\dagger$ remains at all times classically correlated between the system and its environment, displaying neither quantum discord[43,44] nor entanglement, then the reduced system dynamics $\rho_S(t) = \mathcal{E}_t\{\rho_S(0)\} = \mathrm{Tr}_E[\rho_T(t)]$ can be described by a time-independent HE equipped with a legitimate (i.e., non-negative and normalized to unity) probability distribution. Moreover, such ensemble description under classical environments in the absence of back-action has also been discussed in the literature[45–47].

However, given exclusively the knowledge on the reduced system dynamics $\mathcal{E}_t$, it is impossible to fully verify the correlations between the system and its environment. Counter-intuitively, even if we have limited access to the system alone, the emergence of nonclassical correlations can be witnessed, whenever one has no way to simulate the dynamics with any HEs equipped with a legitimate probability distribution. Such impossibility to simulate arises from the buildup of nonclassical correlations. On the other hand, if such simulation is possible, one can explain $\mathcal{E}_t$ as a classical random rotation model. We, therefore, define the negative values of the quasi-distribution $\wp_\lambda$ within the simulating HE as an indicator of process nonclassicality[17].

**Existence and uniqueness of the CHER for pure dephasing**. Here we promote the $\wp_\lambda$ within the simulating HE as a CHER for a reduced system dynamics. In particular, by further investigating the underlying algebraic structures, we can show that such CHER for pure dephasing is even faithful, provided diagonal member Hamiltonians. More precisely, for any pure dephasing dynamics, there always exists a unique simulating HE of diagonal member Hamiltonians, equipped with either a legitimate or quasi-distribution.

The proof will become intelligible only after introducing our procedure to find the CHER below. We postpone it to Supplementary Note 8.

Since $\wp_\lambda$ is a distribution function over the parameter space of diagonal member Hamiltonians, along with the Fourier transform on the group $\mathcal{G}$ in Eq. (3), this endows the CHER with a geometric interpretation of pure dephasing in terms of random rotation model. Consequently, the CHER is particularly competent in characterization of the nonclassicality of pure dephasing.

**The nonclassicality measure for pure dephasing dynamics**. Having characterized the HE simulation of pure dephasing and its representation, we are now ready to propose the measure of nonclassicality of dynamics. The measure aims to provide an operational quantification on the nonclassicality of a pure dephasing dynamics. Due to the existence and uniqueness, every pure dephasing $\mathcal{E}_t$ can be assigned a unique (quasi-)distribution $\wp_\lambda$. We emphasize that it is the distribution $\wp_\lambda$ which gives the characterization of the nonclassicality: unless they correspond to legitimate probabilities, no HE exists for the exact simulation.

The nonclassicality measure for a dynamics $\mathcal{E}_t$ assigned with a unique (quasi-)distribution $\wp_\lambda$ is as follows,

$$\begin{aligned}\mathcal{N}\{\mathcal{E}_t\} &= \inf_{p_\lambda} D(\wp_\lambda, p_\lambda), \text{ with } D(p_\lambda, p'_\lambda) \\ &= \int_{\mathcal{G}} \tfrac{1}{2} |p_\lambda - p'_\lambda| d\lambda,\end{aligned} \tag{4}$$

where the infimum runs overall classical probability distributions $p_\lambda$ over the parameter space $\mathcal{G}$ of the diagonal member Hamiltonians. The variational distance $D(p_\lambda, p'_\lambda)$ has an operational meaning as the single-shot distinguishability: it quantifies the highest success probability of distinguishing two probabilistic systems $p_\lambda$ and $p'_\lambda$, such that $p_{\text{success}} = [1 + D(p_\lambda, p'_\lambda)]/2$.

The measure proposed in Eq. (4) contains advantages and useful properties for the quantification. First, the measure has a clear operational meaning. It tells how well a dynamics $\mathcal{E}_t$ can be simulated by a HE. The possibility of making success or failure in the simulation with a HE can be found. Second, the measure is monotonic that the larger it is, the harder a classical simulation is. This follows from the fact that the classical dynamics of pure dephasing forms a convex set, i.e., their probabilistic mixture is also classical. The proof is presented in Supplementary Note 3. It is noteworthy that the convexity can be constructed by considering (quasi-)probabilities of dynamics, but not dynamics

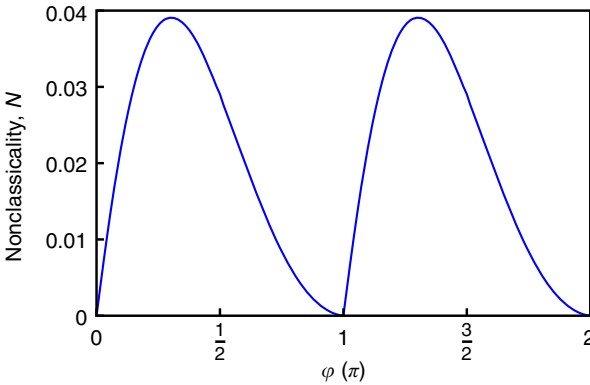

**Fig. 1** The nonclassicality of the qubit pure dephasing. We consider the qubit pure dephasing reduced from the extended spin-boson model, wherein $\varphi$ (in unit of $\pi$) is the relative phase between the coupling constants of the qubit-pair to the common boson environment. The nonclassicality $\mathcal{N}$ is quantified according to Eq. (4). In this example, the Ohmic spectral density $\mathcal{J}(\omega) = \omega exp(-\omega/\omega_c)$ with cut-off $\omega_c = 1$ and the zero-temperature limit are considered

*per se*. Finally, we also note that the measure shares some similarities with the quantification of non-Markovianity[48].

In what follows, we consider the nonclassicality of pure dephasing dynamics on a single qubit reduced from the extended spin-boson model[17] with a relative phase between the coupling constants, i.e., $g_{2,\mathbf{k}} = g_{1,\mathbf{k}}e^{i\varphi}$. The quasi-distribution $\wp_{o1}^{(X)}(\omega)$ represents the single qubit pure dephasing and, consequently, its nonclassicality varies with $\varphi$. The results are shown in Fig. 1.

**Retrieval of the (quasi-)distribution**. Given a HE, it is, in principle, straightforward to calculate the averaged dynamics of an isolated system, according to Eq. (1) [or, equivalently, to Eq. (3)]. Nevertheless, to find the solution to the inverse transform of Eq. (3), i.e., retrieval of the (quasi-)distribution within the simulating HE for a given reduced dynamics, is formidable in general, in contrast to the conventional inverse Fourier transform. Consequently, to establish a systematic procedure to find the CHER of pure dephasing dynamics is very desirable.

In view of the qubit pure dephasing in Eq. (2), to simulate any higher dimensional pure dephasing dynamics, we focus on the traceless and diagonal member Hamiltonian such that $\widehat{H}_\lambda = \lambda\widehat{L}$ belongs to the Cartan subalgebra (CSA) $\mathfrak{H}$ of $\mathfrak{su}(n)$ (see Methods). The tracelessness is due to the fact that the trace plays no role in describing the dynamics. Additionally, since the adjoint representation preserves the structure of commutator, the adjoint representation of $\mathfrak{H}$ is also a CSA of $\mathfrak{sl}(\mathfrak{u}(n))$. We, therefore, have the following commutativity $[\lambda\widehat{L}, \lambda'\widehat{L}] = 0 \Leftrightarrow [\lambda\widetilde{L}, \lambda'\widetilde{L}] = 0$.

It should be noted that, even if $\lambda\widehat{L} \in \mathfrak{H}$ can be chosen to be diagonal, $\lambda\widetilde{L}$ itself may not necessarily be diagonal as well since the generators of $\mathfrak{u}(n)$ are not the suitable bases for diagonalizing it. As we will see below, the diagonalization of the adjoint representation is a critical step to the retrieval of the (quasi-)distribution for pure dephasing.

Furthermore, the conventional inverse Fourier transform does not work because we are now dealing with linear maps in the $\mathfrak{sl}(\mathfrak{u}(n))$ space. To efficiently establish a set of equations governing the CHER of pure dephasing, we inevitably encounter increasingly many mathematical terminologies, especially those specifying the intrinsic algebraic structures within the CHER. To make our procedure transparent, we instead demonstrate several

examples, each of which reveals the central concepts of our procedure, rather than elaborate the mathematical tutorial. Our approach can be easily generalized to higher dimensional pure dephasing.

**Procedure towards the CHER of pure dephasing**. We begin with the case of qubit pure dephasing. Although this problem has been discussed[17], it relies on the conventional Fourier transform and Bochner's theorem[49] and cannot be generalized to higher dimensional systems. Here we recast it into Eq. (3). This helps us to establish a systematic procedure for higher dimensional problems.

Within a properly chosen basis, a qubit pure dephasing, reduced from a system–environment arrangement, can be expressed in the same form as Eq. (2). Unlike the one resulting from ensemble average, the dephasing factor $\phi(t) = exp[-i\theta(t) - \Phi(t)]$ is determined by the system–environment interaction, where $\theta(t)$ ($\Phi(t)$) is a real odd (even) function on time $t$, respectively, such that $\phi(0) = 1$, $|\phi(t)| \leq 1$, and $\phi(-t) = \phi^*(t)$. The dynamical linear map $\mathcal{E}_t^{(\tilde{\sigma})}$ can be constructed by applying $\mathcal{E}_t\{\hat{\sigma}_m\} = \sum_{l=0}^3 \hat{\sigma}_l[\mathcal{E}_t^{(\tilde{\sigma})}]_{lm}$ on each generator, where $\hat{\sigma}_0 = \hat{I}$ is the identity and $\hat{\sigma}_{1,2,3}$ denotes the three Pauli matrices.

To find the CHER, we mean to find a (quasi-)distribution $\wp(\omega)$ encapsulated within the simulating HE $\{(\wp(\omega), \widehat{H}_\omega = \omega\hat{\sigma}_z/2)\}_\omega$ satisfying

$$\mathcal{E}_t^{(\tilde{\sigma})} = \int_\mathbb{R} \wp(\omega)e^{-i(\omega\tilde{\sigma}_z/2)t}d\omega. \qquad (5)$$

The same conclusion $exp[-i\theta(t) - \Phi(t)] = \int_\mathbb{R} \wp(\omega)e^{-i\omega t}d\omega$ is easily seen after diagonalizing Eq. (5) (see Supplementary Note 4 for more details). Finally, performing the conventional inverse Fourier transform leads to the desired result $\wp(\omega)$.

To understand the deeper insight behind the diagonalization, we observe that the diagonalization changes the basis from the three pauli matrices into raising and lowering operators and leaves $\hat{\sigma}_z$ invariant; namely, $\{\hat{\sigma}_+, \hat{\sigma}_-, \hat{\sigma}_z\}$, which are the generators of $\mathfrak{sl}(2)$. In other words, they are the common "eigenvectors" of $\widehat{H}_\omega$ with "eigenvalues" $\pm 1$ in the sense of the adjoint representation, $\widetilde{H}_\omega(\hat{\sigma}_\pm) = [\omega\hat{\sigma}_z/2, \hat{\sigma}_\pm] = \pm 1 \cdot \omega\hat{\sigma}_\pm$ (see Supplementary Note 5 for more details). The eigenvalues $\pm 1$ are referred to as the roots (denoted by $\alpha_{1,2}$) associated to the root spaces span$\{\hat{\sigma}_\pm\}$, spanned by the operators $\hat{\sigma}_\pm$, respectively. However, for higher dimensional systems, the roots are no longer real scalars but vectors in an Euclidean space. This can be better seen as follow.

A qutrit pure dephasing can be written as

$$\rho(t) = \mathcal{E}_t\{\rho_0\} = \begin{bmatrix} \rho_{11} & \rho_{12}\phi_1(t) & \rho_{13}\phi_4(t) \\ \rho_{21}\phi_2(t) & \rho_{22} & \rho_{23}\phi_6(t) \\ \rho_{31}\phi_5(t) & \rho_{32}\phi_7(t) & \rho_{33} \end{bmatrix}, \qquad (6)$$

To guarantee the Hermicity of $\rho(t)$, the dephasing factors must further satisfy $\phi_1(t) = \phi_2^*(t)$, and so on.

To expand $\rho$ as a nine-dimensional column vector, it is natural to use the Gell-Mann matrices (denoted by $\hat{\sigma}_m$, $m = 1, ..., 8$) as the generators of $\mathfrak{su}(3)$. However, after the diagonalization, the basis is changed into that of $\mathfrak{gl}(3)$ (e.g., $\widehat{K}_0 = \hat{I}$, $\widehat{K}_1 = \widehat{K}_2^\dagger = (\hat{\sigma}_1 + i\hat{\sigma}_2)/2$, and $\widehat{K}_3 = \widehat{L}_3 = \hat{\sigma}_3$). Within this basis, the dynamical linear map $\mathcal{E}_t^{(\tilde{L})}$ is diagonalized, i.e., $\mathcal{E}_t\{\widehat{K}_m\} = \widehat{K}_m\phi_m(t)$.

In this case, we consider the member Hamiltonian $\widehat{H}_\lambda = (\lambda_3\widehat{L}_3 + \lambda_8\widehat{L}_8)/2 \in \mathfrak{H}$ and $\lambda = (\lambda_3, \lambda_8) \in \mathbb{R}^2$. After estimating all the commutators $[\widehat{H}_\lambda, \widehat{K}_m] = (\boldsymbol{\alpha}_m \cdot \boldsymbol{\lambda})\widehat{K}_m$, we obtain its adjoint

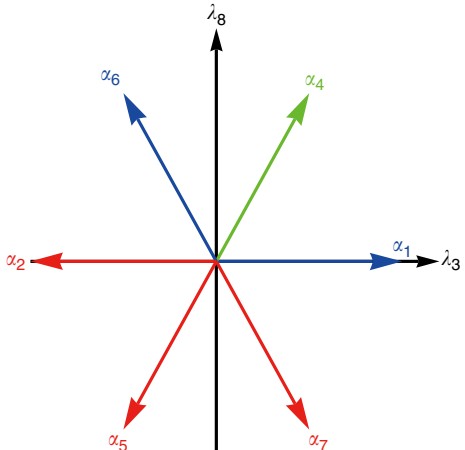

**Fig. 2** The root system R of $\mathfrak{su}(3)$. It consists of six non-zero root vectors on the $\lambda_3 - \lambda_8$ plane. Among them, $\boldsymbol{\alpha}_1$ (blue), $\boldsymbol{\alpha}_4$ (green), and $\boldsymbol{\alpha}_6$ (blue) are positive and the other three (red) are negative since roots are always come in pair with opposite directions. Also, $\boldsymbol{\alpha}_1$ and $\boldsymbol{\alpha}_6$ are simple because $\boldsymbol{\alpha}_4 = \boldsymbol{\alpha}_1 + \boldsymbol{\alpha}_6$ is a combination of simple roots

representation $\widetilde{H}_{\boldsymbol{\lambda}} = \left(\lambda_3\widetilde{L}_3 + \lambda_8\widetilde{L}_8\right)/2$, which is diagonal in the $\mathfrak{gl}(3)$ basis.

Finally, according to Eq. (3) $\mathcal{E}_t^{(\widetilde{L})} = \int_{\mathbb{R}^2} \wp(\lambda_3, \lambda_8)e^{-i\widetilde{H}_{\boldsymbol{\lambda}}t}d\lambda_3 d\lambda_8$, we conclude that the (quasi-)distribution $\wp(\lambda_3, \lambda_8)$ is governed by the following simultaneous Fourier transforms:

$$\phi_1(t) = \int_{\mathbb{R}^2} \wp(\lambda_3, \lambda_8)e^{-i(\boldsymbol{\alpha}_1\cdot\boldsymbol{\lambda})t}d\lambda_3 d\lambda_8, \tag{7}$$

$$\phi_4(t) = \int_{\mathbb{R}^2} \wp(\lambda_3, \lambda_8)e^{-i(\boldsymbol{\alpha}_4\cdot\boldsymbol{\lambda})t}d\lambda_3 d\lambda_8, \tag{8}$$

$$\phi_6(t) = \int_{\mathbb{R}^2} \wp(\lambda_3, \lambda_8)e^{-i(\boldsymbol{\alpha}_6\cdot\boldsymbol{\lambda})t}d\lambda_3 d\lambda_8. \tag{9}$$

We can collect the six non-zero root vectors $\boldsymbol{\alpha}_m$. They are two-dimensional vectors of equal length in the $\lambda_3$-$\lambda_8$ plane forming the root system R of $\mathfrak{su}(3)$. We plot them in Fig. 2. Further details are given in Supplementary Note 6.

Similarly, for $n$-dimensional pure dephasing, each member Hamiltonian $\widehat{H}_{\boldsymbol{\lambda}}$, taken from the $\mathfrak{H}$ of $\mathfrak{su}(n)$, possesses $n-1$ free parameters $\boldsymbol{\lambda} = \{\lambda_{k^2-1}\}_{k=2,3,\dots,n}$; meanwhile, the (quasi-)distribution $\wp(\boldsymbol{\lambda})$ is defined on the $(n-1)$-dimensional Euclidean space. Moreover, the action of $\widehat{H}_{\boldsymbol{\lambda}}$ on the $n^2 - n$ root spaces span$\{\widehat{K}_m\}$ is described by the root system R = $\{\boldsymbol{\alpha}_m\}_m$, consisting of $n^2 - n$ real vectors of $(n-1)$-dimension. Further properties of R reduce the complexity of our procedure (see Methods).

Consequently, combining the techniques, i.e., the adjoint representation, the Fourier transform on groups, and the root space decomposition, we can concisely formulate our procedure to find the CHER $\wp(\boldsymbol{\lambda})$ for the $n$-dimensional pure dephasing. We restrict ourselves to the diagonal member Hamiltonians (in $\mathfrak{H}$) and establish its root system R. The (quasi-)distribution $\wp(\boldsymbol{\lambda})$ is characterized by the $(n^2 - n)/2$ Fourier transforms with respect to positive roots and its corresponding dephasing factor $\phi_m(t)$ associated to the root space span$\{\widehat{K}_m\}$:

$$\phi_m(t) = \int_{\mathbb{R}^{n-1}} \wp(\boldsymbol{\lambda})e^{-i(\boldsymbol{\alpha}_m\cdot\boldsymbol{\lambda})t}d^{n-1}\boldsymbol{\lambda}, \text{ for positive roots } \boldsymbol{\alpha}_m. \tag{10}$$

Furthermore, the simple roots define a new set of random variables $x_m = \boldsymbol{\alpha}_m\cdot\boldsymbol{\lambda}$, for simple roots $\boldsymbol{\alpha}_m$, and their corresponding equations define the marginals of $\wp(\boldsymbol{\lambda})$ along $x_m$. The other equations describe the correlations among $x_m$.

**Example: qubit pair pure dephasing.** As an instructive paradigm demonstrating our procedure to find the CHER of pure dephasing, we consider the extended spin-boson model consisting of a non-interacting qubit pair coupled to a common boson bath (Fig. 3a) with total Hamiltonian $\widehat{H}_T = \sum_{j=1,2} \omega_j\hat{\sigma}_{z,j}/2 + \sum_{\mathbf{k}} \omega_{\mathbf{k}}\hat{b}_{\mathbf{k}}^{\dagger}\hat{b}_{\mathbf{k}} + \sum_{j,\mathbf{k}} \hat{\sigma}_{z,j} \otimes \left(g_{j,\mathbf{k}}\hat{b}_{\mathbf{k}}^{\dagger} + g_{j,\mathbf{k}}^*\hat{b}_{\mathbf{k}}\right)$. We now focus on the pure dephasing of the qubit pair as a $4 \times 4$ system. The full dynamics has been given in ref. [17].

To simulate the qubit pair pure dephasing, the diagonal member Hamiltonian is taken from the $\mathfrak{H}$ of $\mathfrak{su}(4)$ $\widehat{H}_{\boldsymbol{\lambda}} = \left(\lambda_3\widehat{L}_3 + \lambda_8\widehat{L}_8 + \lambda_{15}\widehat{L}_{15}\right)/2$ and $\wp(\boldsymbol{\lambda})$ is a (quasi-)distribution over $\mathbb{R}^3$ space with $\lambda_3$, $\lambda_8$, and $\lambda_{15}$ being its axes. Note that the $\mathfrak{su}(4)$ has six positive root vectors and three among them are simple, and all positive root vectors can be obtained by combining simple ones (Fig. 3b). We perform the change of variables $x_m = \boldsymbol{\alpha}_m\cdot\boldsymbol{\lambda}$, $m = 1, 6, 13$. Then, the (quasi-)distribution changes as $\wp(\boldsymbol{\lambda}) \mapsto \wp'(x_1, x_6, x_{13})$. The three axes of $\wp'(\mathbf{x})$ are defined by the three simple root vectors.

Additionally, since $\phi_6(t) = 1$, by observing the special correspondences between root vectors and dephasing factors, we can assume that

$$\wp'(\mathbf{x}) = \wp_6(x_6)\wp_{1,13}(x_1, x_{13}) \tag{11}$$

is separable into two parties. The Fourier equation for $\phi_6(t)$ leads to the result that $\wp_6(x_6) = \delta(x_6)$ and those for $\phi_1(t)$ and $\phi_{13}(t)$ specify the marginals of $\wp_{1,13}(x_1, x_{13})$ along the direction $\boldsymbol{\alpha}_1$ and $\boldsymbol{\alpha}_{13}$, respectively; meanwhile the one for $\phi_9(t)$

$$\phi_9(t) = \int_{\mathbb{R}^2} \wp_{1,13}(x_1, x_{13})e^{-ix_1 t}e^{-ix_{13}t}dx_1 dx_{13} \tag{12}$$

describes the correlation between $x_1$ and $x_{13}$.

For the case of Ohmic spectral density $\mathcal{J}(\omega) = \omega exp(-\omega/\omega_c)$ in the zero-temperature limit, Eq. (12) can be recast into a conventional two-dimensional Fourier transform by a simple ansatz. Then, $\wp_{1,13}(x_1, x_{13})$ can be easily obtained by conventional inverse transform and the numerical result is shown in Fig. 3c. It exhibits manifest negative regions and illustrates the nonclassical nature of the qubit pair pure dephasing. Detailed calculations are given in Supplementary Note 7.

Finally, having introducing our procedure to find the CHER, we combine it with the investigation on the intrinsic algebraic structure. Then the uniqueness of the CHER for pure dephasing is intelligible and the detailed proof is given in Supplementary Note 8.

It is worthwhile to recall that similar models, in which several qubits were coupled identically to a common bath, had been considered[50–52], wherein the suppression of decoherence within certain Hilbert subspace had been discovered. These studies spurred the development of the theory of decoherence-free-subspace[53,54], which is conceived as a promising solution to circumvent the obstacle of decoherence in quantum information science. The phenomenon of coherence-preserving can be observed in our paradigm as well and is related to the delta component $\wp_6(x_6) = \delta(x_6)$ on $x_6$. Consequently, our procedure provides a potential application in the detection of decoherence-free-subspace in terms of delta components in the (quasi-)distribution.

**Proposed experimental realization.** Finally, to underpin the practical feasibility of our approach, here we explain how to recover the dynamical linear map $\mathcal{E}_t^{(\widetilde{L})}$ from the measurable $\chi$ matrix, which is a typical way to characterize arbitrary dynamics.

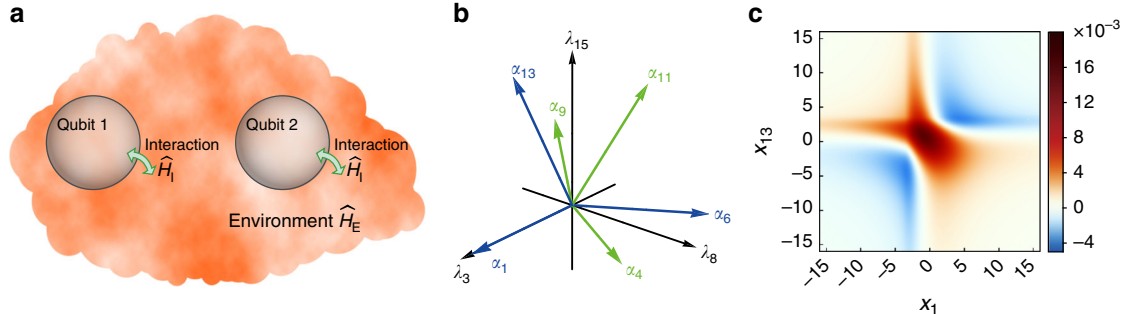

**Fig. 3** Nonclassicality of the qubit pair pure dephasing. **a** A schematic illustration of our extended spin-boson model, describing a pair of non-interacting qubits coupled to a common boson environment. **b** To simulate the qubit pair pure dephasing, $\wp(\boldsymbol{\lambda})$ is a (quasi-)distribution over $\mathbb{R}^3$ space spanned by $\lambda_3$, $\lambda_8$, and $\lambda_{15}$. Here we show the six positive root vectors of $\mathfrak{su}(4)$. Three simple root vectors (blue) define a new set of random variables. The other three non-simple root vectors (green) can be expressed as a combination of simple ones, e.g., $\boldsymbol{\alpha}_9 = \boldsymbol{\alpha}_1 + \boldsymbol{\alpha}_6 + \boldsymbol{\alpha}_{13}$. **c** The function $\wp_{1,13}(x_1, x_{13})$ distributes over the plane spanned by $x_1$ and $x_{13}$. For the case of Ohmic spectral density in the zero-temperature limit and $\omega_c = 1$, it shows manifest negative regions and therefore indicates the nonclassicality of the qubit pair pure dephasing

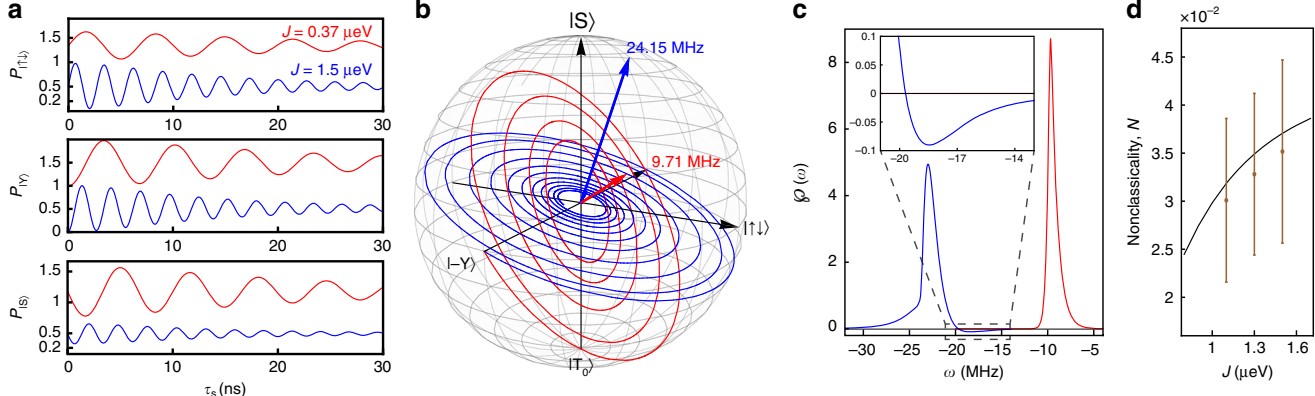

**Fig. 4** Numerical simulation of the S-T$_0$ qubit pure dephasing. **a** The return probabilities $P_{|j\rangle}(\tau_s)$ are measured by projecting the states onto each axis after a free induction decay time $\tau_s$. Here, we show two numerical simulations at different $J$ values. **b** The trajectories can be depicted in the Bloch sphere and the dynamics are therefore clearly visualized. The axes of rotation, as well as the angle $\Omega$ between the $|S\rangle$-axis, are identified by the normal vectors. **c** According to the rotation axes identified in (**b**), a unitary rotation $\hat{R}_\Omega$ recovers the standard form in Eq. (2). Then our procedure is applicable. The resulting $\wp(\omega)$'s reflect several physical intuitions, as explained in the main text. **d** The corresponding nonclassicality $\mathcal{N}$ at different $J$ can be estimated according to Eq. (4). It increases with $J$ in our simulation. In line with a realistic experimental modeling, statistically fluctuating noise is taken in account. The average nonclassicalities (brown dots) are reduced due to the noise. The brown error bars are the standard deviations of the series of nonclassicality $\mathcal{N}$ values obtained by repeatedly performing the noise simulation. More details are given in Supplementary Note 9

The matrix elements $\chi_{l,m}(t)$ are measured following the quantum process tomography technique, which has been applied in various architectures, e.g., optics[55–57], trapped ions[58,59], and superconductors[60,61].

Note that $\mathcal{E}_t^{(L)}$ on the left-hand side of Eq. (3) describes the complete time evolution of the system, i.e., we need to generate raw data of $\chi_{l,m}(t)$ as a time sequence. While this implies repeating the experiment for different time intervals, it does in principle not impose additional technical difficulty. Finally, $\mathcal{E}_t^{(\widetilde{L})}$ can be reconstructed by combining the measured $\chi_{l,m}(t)$ (see Methods).

Here, we also demonstrate a numerical simulation of the quantum state tomography experiment in the S-T$_0$ qubit[24,25]. With spin relaxation on the order of milliseconds[62], the qubit dynamics is well approximated as pure dephasing on the time scale of tens of nanoseconds. The qubit state is detected by measuring the return probabilities, i.e., projective measurements onto each axis of the Bloch sphere, after a free induction decay time $\tau_s$, as shown in Fig. 4a (see Methods). With the measured return probabilities, we can depict the trajectories in the Bloch sphere (Fig. 4b). This allows

us to fully reconstruct the dynamics $\varepsilon_t$ of the qubit. Then applying our procedure outlined above, we can obtain the resulting $\wp(\omega)$ shown in Fig. 4c. They reflect the fact that the $|S\rangle$ possesses a lower eigenenergy than $|T_0\rangle$, and the physical intuition that the shorter the coherence time, the broader the $\wp(\omega)$. Having recovered $\wp(\omega)$, the nonclassicality values can be estimated according to Eq. (4). To achieve realistic experimental conditions, we dress the theoretical model with statistical fluctuations (Fig. 4d). This confirms the robustness of the nonclassicality detection against experimental errors (see Supplementary Note 9).

## Discussion

The studies on unveiling genuine quantum properties are very important since these discover the fundamental principle of nature and spur the growth of different branches in physics and technologies. Particularly, in the field of quantum information science, highly quantum-correlated systems are critical resources for prominent quantum information tasks which can hardly be accomplished efficiently by classical computers.

By genuine quantum properties, we refer to those that can never be resembled by classical strategies. For example, Bell's

inequality is derived based on the assumption of realism and locality, while the Wigner function explain a boson field in terms of classical phase space. Inspired by these works, our characterization of process nonclassicality stems from the correspondence between the averaged dynamics of a HE and the dynamics reduced from a system–environment arrangement[17].

By introducing the CHER, the role of classical strategy played by the simulating HE for a dynamics is even more apparent. The (quasi-)distribution is endowed with an explanation in terms of a random rotation model. This also implies that the nonclassical properties of a dynamics can be well-characterized by a (quasi-) distribution.

Our main achievement here lies in the establishment of a constructive procedure to retrieve the (quasi-)distributions for pure dephasing of any dimension. Additionally, along with the analysis of the underlying algebraic structure, we also achieve to prove its existence and uniqueness provided commuting member Hamiltonians. Therefore, the CHER of pure dephasing is faithful. Accordingly, based on our studies, we propose a measure of nonclassicality of pure dephasing by comparing the (quasi-)distributions in terms of variational distance. We also show that our measure is reasonable due to its convexity.

In order to make our procedure viable, we discuss how to implement our approach with the raw data measured by quantum process tomography. Furthermore, we also demonstrate a numerical simulation of the S-$T_0$ qubit quantum state tomography, with which we implement our approach step by step.

Finally, let us remark that the generalization to the cases beyond pure dephasing or even nonunital dynamics invokes nonabelian algebraic structures. The Baker–Campbell–Hausdorff formula is then required and therefore complicates the formulation here. On the other hand, our approach highlights an inherent difference between dephasing and dissipative dynamics in terms of their underlying algebras. This may provide a new route toward the theory of open systems. Additionally, we also find that it would be interesting to investigate how the notion of dynamical process nonclassicality is related to other quasi-distributions[63].

## Methods

**Adjoint representation**. The adjoint representation is a particularly important tool in the theory of Lie algebra. It assigns each element in a Lie algebra $\mathfrak{L}$ an endomorphism in $\mathfrak{gl}(\mathfrak{L})$ (i.e., a homomorphism from $\mathfrak{L}$ to itself) in terms of Lie bracket. Therefore, $\mathfrak{gl}(\mathfrak{L})$ is a Lie algebra consisting of linear maps acting on $\mathfrak{L}$, wherein $\mathfrak{L}$ plays the role of a vector space with the generators being its basis. The adjoint representation of each generator is constructed in terms of structure constants $c_{klm}$. See Supplementary Note 1 for more details.

**Cartan subalgebra**. The structure of a Lie algebra $\mathfrak{L}$ is largely determined by its Lie bracket, i.e., the commutator acting on $\mathfrak{L}$. A Lie algebra is said to be abelian if all its elements are mutually commutative. Let $\mathfrak{H}$ be a Lie subalgebra of $\mathfrak{L}$. $\mathfrak{H}$ is said to be the CSA of $\mathfrak{L}$ if $\mathfrak{H}$ is the maximal abelian (and semisimple) subalgebra. A very important property is that, for a Lie algebra consists of matrices, the elements in its CSA are all simultaneously diagonalizable for a suitably chosen basis.

In our case, to simulate pure dephasing dynamics, we deal with traceless and diagonal member Hamiltonians, taken from $\mathfrak{H}$ of $\mathfrak{su}(n)$. To be noted, since the adjoint representation preserves the Lie bracket, the adjoint representation of $\mathfrak{H}$ is also a CSA of $\mathfrak{sl}(\mathfrak{u}(n))$. However, even if $\mathfrak{H}$ is diagonal, its adjoint representation may not necessarily be diagonal as well since the generators of $\mathfrak{u}(n)$ are not the suitable basis for diagonalizing it.

**Root system**. For $n$-dimensional systems, there are $(n-1)$ generators in the $\mathfrak{H}$ of $\mathfrak{su}(n)$. Therefore, each member Hamiltonian possesses $(n-1)$ parameters $\widehat{H}_\lambda = \sum_{k=2}^{n} \lambda_{k^2-1} \widehat{L}_{k^2-1}/2$, with $\{\widehat{L}_{k^2-1}\}_{k=2,3,\ldots,n}$ being the generators of $\mathfrak{H}$. Additionally, the $(n^2-n)$ roots $\boldsymbol{\alpha}_m$, associated to each root space span$\{\widehat{K}_m\}$, are $(n-1)$-dimensional vectors, forming the root system R $= \{\boldsymbol{\alpha}_m\}_m$ of $\mathfrak{su}(n)$. Besides, according to the theory of root space decomposition, the root system possesses the following critical properties: (1) the roots come in pairs in the sense that, if $\boldsymbol{\alpha}_m$ is a root, then $-\boldsymbol{\alpha}_m$ is a root as well. This reduces the number of equations half since we are sufficient to consider the positive roots alone. (2) Among the $(n^2-n)/2$ positive roots, $(n-1)$ simple roots provide the marginal of $\wp$ along different

directions and the others provide the information on the correlations between them. (3) For $\mathfrak{su}(n)$, the angle between any two non-pairing roots must be either $\pi/3$, $\pi/2$, or $2\pi/3$. Furthermore, with the Fourier transform on groups, an $n$-dimensional pure dephasing is characterized by $(n^2-n)/2$ complex functions $\phi_m(t)$, which are the dephasing factors associated to each root space span$\{\widehat{K}_m\}$.

**Reconstructing $\mathcal{E}_t^{(\widetilde{L})}$ from the $\chi$ matrix**. In our approach, the reduced system dynamics is fully characterized by the dynamical linear map $\mathcal{E}_t^{(\widetilde{L})}$, which is an $n^2 \times n^2$ matrix acting on a state column vector $\rho = \{n^{-1}, \boldsymbol{\rho}\} \in \mathbb{R}^{n^2}$. On the other hand, in a quantum process tomography experiment, the dynamics is characterized by the measurable $\chi$ matrix representation, with the matrix elements defined according to

$$\mathcal{E}_t\{\rho_0\} = \sum_{l,m=0}^{n^2-1} \chi_{l,m}(t) \widehat{L}_l \rho_0 \widehat{L}_m. \tag{13}$$

Note that we have used the Hermiticity $\widehat{L}_m^\dagger = \widehat{L}_m$ in the above expression.

Now we demonstrate how to reconstruct $\mathcal{E}_t^{(\widetilde{L})}$ from the measured $\chi_{l,m}(t)$. For a given dynamics $\varepsilon_t$, the matrix elements $\left[\mathcal{E}_t^{(\widetilde{L})}\right]_{jk}$ are defined by applying

$$\mathcal{E}_t\{\widehat{L}_k\} = \sum_{j=0}^{n^2-1} \widehat{L}_j \left[\mathcal{E}_t^{(\widetilde{L})}\right]_{jk} \tag{14}$$

on each generator $\widehat{L}_k$. On the other hand, according to the measured $\chi_{l,m}(t)$ in Eq. (13), we have

$$\mathcal{E}_t\{\widehat{L}_k\} = \sum_{l,m=0}^{n^2-1} \chi_{l,m}(t) \widehat{L}_l \widehat{L}_k \widehat{L}_m. \tag{15}$$

From the above two equations, we can deduce that

$$\left[\mathcal{E}_t^{(\widetilde{L})}\right]_{jk} = \frac{1}{2} \sum_{l,m=0}^{n^2-1} \chi_{l,m}(t) \text{Tr} \widehat{L}_j \widehat{L}_l \widehat{L}_k \widehat{L}_m, j, k \neq 0, \tag{16}$$

$$\left[\mathcal{E}_t^{(\widetilde{L})}\right]_{j0} = \frac{1}{2} \sum_{l,m=0}^{n^2-1} \chi_{l,m}(t) \text{Tr} \widehat{L}_j \widehat{L}_l \widehat{L}_m, j \neq 0, \tag{17}$$

$$\left[\mathcal{E}_t^{(\widetilde{L})}\right]_{0k} = \frac{1}{n} \sum_{l,m=0}^{n^2-1} \chi_{l,m}(t) \text{Tr} \widehat{L}_l \widehat{L}_k \widehat{L}_m, k \neq 0, \tag{18}$$

and

$$\left[\mathcal{E}_t^{(\widetilde{L})}\right]_{00} = \chi_{0,0}(t) + \frac{2}{n} \sum_{l=1}^{n^2-1} \chi_{l,l}(t). \tag{19}$$

In the above equations, we have used the facts that $\widehat{L}_0 = \hat{I}$ and $\text{Tr} \widehat{L}_j^2 = 2$ for $j \neq 0$.

**Recovering the S-$T_0$ trajectory from measured data**. For a double-quantum-dot S-$T_0$ qubit, the three axes of the Bloch sphere are conventionally defined as $|X\rangle = (|S\rangle + |T_0\rangle)/\sqrt{2} = |\uparrow\downarrow\rangle$, $|Y\rangle = (|S\rangle - i|T_0\rangle)/\sqrt{2}$, and $|Z\rangle = |S\rangle = (|\uparrow\downarrow\rangle - |\downarrow\uparrow\rangle)/\sqrt{2}$ being the singlet state, as shown in Fig. 4b. The free Hamiltonian in the S-$T_0$ basis is

$$\widehat{H}_{\text{S}T_0} = \begin{bmatrix} -J & g\mu_B \Delta B_{\text{nuc}}^z \\ g\mu_B \Delta B_{\text{nuc}}^z & 0 \end{bmatrix}, \tag{20}$$

where $J = 0.37\,\mu\text{eV}$ (red) and $1.5\,\mu\text{eV}$ (blue) is the exchange energy between two dots, $\Delta B_{\text{nuc}}^z = 10.5$ mT is the hyperfine field gradient, $g = -0.44$ is the $g$-factor for GaAs, and $\mu_B = 57.8\,\mu\text{eVT}^{-1}$ is Bohr's magneton. Various kinds of initial states can be prepared with carefully designed pulse by controlling the voltage detuning between the quantum dots. After the initialization, the qubit undergoes a free induction decay for a time period $\tau_s$. Finally, projective measurements onto each axis are performed.

We numerically simulate the return probabilities $P_{|\uparrow\downarrow\rangle}(\tau_s)$, $P_{|Y\rangle}(\tau_s)$, and $P_{|S\rangle}(\tau_s)$ to each axis (Fig. 4a). Then the density matrix $\rho(\tau_s) = \left[\hat{I} + \sum_{j=X,Y,Z} r_j(\tau_s)\hat{\sigma}_j\right]/2$ can be determined by

$$r_j(\tau_s) = 2P_{|j\rangle}(\tau_s) - 1, j = X, Y, Z. \tag{21}$$

And one can depict the trajectory $\boldsymbol{r}(\tau_s) = \{r_X(\tau_s), r_Y(\tau_s), r_Z(\tau_s)\}$ in the Bloch sphere (Fig. 4b). This helps us to identify the axis of rotation with bare rotation frequencies $\omega = \sqrt{J^2 + \left(2g\mu_B \Delta B_{\text{nuc}}^z\right)^2}/\hbar$ and the angle $\Omega$ between the $|S\rangle$-axis.

Finally, a unitary rotation $\widehat{R}_\Omega \rho(\tau_s) \widehat{R}_\Omega^\dagger$ with $\widehat{R}_\Omega = exp[i\Omega\tilde{\sigma}_Y/2]$ recover the standard form in Eq. (2). Our procedure is then applicable and leads to

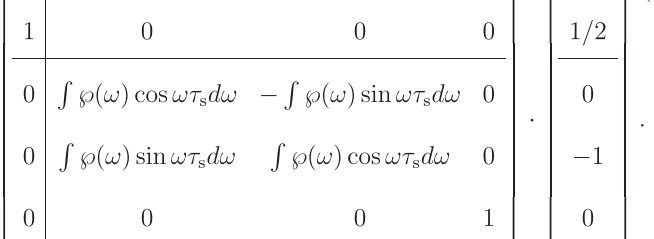

$$
\begin{bmatrix}
1/2 \\
(r_X(\tau_s)\cos\Omega - r_Z(\tau_s)\sin\Omega)/2 \\
r_Y(\tau_s)/2 \\
(r_X(\tau_s)\sin\Omega + r_Z(\tau_s)\cos\Omega)/2
\end{bmatrix} =
$$

$$
\begin{bmatrix}
1 & 0 & 0 & 0 \\
0 & \int \wp(\omega)\cos\omega\tau_s d\omega & -\int \wp(\omega)\sin\omega\tau_s d\omega & 0 \\
0 & \int \wp(\omega)\sin\omega\tau_s d\omega & \int \wp(\omega)\cos\omega\tau_s d\omega & 0 \\
0 & 0 & 0 & 1
\end{bmatrix} \cdot
\begin{bmatrix}
1/2 \\
0 \\
-1 \\
0
\end{bmatrix}.
\tag{22}
$$

The numerical solutions are shown in Fig. 4c.

Further schematic illustration of the simulation and detailed analysis of the effects of noise are given in Supplementary Note 9.

## Data availability

The data analyzed during the current study are available from the corresponding authors on reasonable request.

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

## Acknowledgements

This work is supported partially by the National Center for Theoretical Sciences and Ministry of Science and Technology, Taiwan, Grants No. MOST 107-2628-M-006-002-MY3, MOST 107-2627-E-006-001, MOST 106-2811-M-006-044, MOST 107-2811-M-006-017, and MOST 107-2811-M-009-527, and Army Research Office (Grant No. W911NF-19-1-0081). J.B. is supported by an Institute of Information and Communications Technology Promotion (IITP) grant funded by the Korean government (MSIP) (Grant No. 2019-0-00831, EQGIS), the KIST Institutional Program (2E29580-19-148), and ITRC Program(IITP2018-2019-0-01402). F.N. is supported in part by the MURI Center for Dynamic Magneto-Optics via the Air Force Office of Scientific Research (AFOSR) (FA9550-14-1-0040), Army Research Office (ARO) (Grant No. W911NF-18-1-0358), Asian Office of Aerospace Research and Development (AOARD) (Grant No. FA2386-18-1-4045), Japan Science and Technology Agency (JST) (Q-LEAP program and CREST Grant No. JPMJCR1676), Japan Society for the Promotion of Science (JSPS) (JSPS-RFBR Grant No. 17-52-50023, and JSPS-FWO Grant No. VS.059.18N), RIKEN-AIST Challenge Research Fund, and the John Templeton Foundation.

## Author contributions

H.-B.C. conceived the research and carried out the calculations, with help from P.-Y.L. and C.G., under the supervision of Y.-N.C. J.B. proposed the idea of variational distance. Y.-N.C. and F.N. were responsible for the integration among different research units. All authors contributed to the discussion of the central ideas and to the manuscript.

## Additional information

**Competing interests:** The authors declare no competing interests.

