## [Peer Review File · Nature Communications]

Reviewers' comments:

Reviewer #1 (Remarks to the Author):

Open system dynamics and decoherence studies are important fields of physics. Within this framework, the manuscript demonstrates how to quantify nonclassicality of dephasing processes in generic manner. Even though there exists a few previous studies on the nonclassicality of quantum processes (cited in the manuscript), and some of the authors themselves have recently studied nonclassicality of the dynamics especially in the association of the type of the generated system-environment correlations (ref. 14), the results presented here are highly non-trivial and display significant results for the studies of open quantum systems. The nonclassicality measure introduced in Eq. (5) is conceptually rather simple. However, to demonstrate that this measure is meaningful and useful in quantifying nonclassicality require elaborate mathematical considerations and calculations, as the material of the manuscript shows, reflecting the nontriviality of the results. Once this has been achieved a general way to quantify nonclassicality of dephasing becomes available.

In broader view, there has been during the recent years large amount of papers quantifying non-Markovianity of the open systems dynamics and elsewhere, e.g., a popular area of quantifying quantum coherence has emerged. I think that the current results fit very well for this background and at the same time opens new directions in the open system studies in addition of the importance of quantifying nonclassicality. Before making a definitive recommendation, I would like the authors to address and comment the following points. I think this would improve the readability of the paper considering the broad and various audiences it may have.

In ref. 14 some of the current authors mention that any nonunitary dynamics is classified as nonclassical. Therefore I suppose that the current manuscript focuses on dephasing (unitary) dynamics which may display classical and nonclassical features. Can the authors elaborate why nonunitary dynamics should always be considered nonclassical and whether the current construction, or a variant of it, could be useful or meaningful also for the nonunitary case?

Suppose that one knows that the dynamics follows dephasing and has the subsequent experimental data based on measurements available. In which way and by using what resources one can then conclude whether this corresponds to classical or nonclassical dynamics, and if the latter, then how much?

Expressed in slightly different way: the manuscript mentions that the nonclassicality measure is developed here in the spirit of Wigner function and its negativity. Now one can reconstruct the Wigner function based on experimental data. How does one reconstruct the probability distribution for the purpose of nonclassicality based on the experimental data only?

The authors justify their results by their generality compared to some earlier results, e.g., ref. 13 deals with single bosonic mode. On the other hand, the considered examples in the current contribution are qubit, qutrit, and qubit-qubit systems, i.e., no continuous variable systems are used as examples here. Can the authors say something more about the applicability of their results, e.g. to harmonic oscillator as open system case?

In example cases, the results demonstrate how the negativity of the used probability distribution shows up when changing certain parameters. However, I do not see results or plots on the decoherence functions and how they behave in different cases. Thereby the question is whether the classicality displays itself in the evolution of the coherences? Or whether nonclassicality of the dynamics can be associated to specific dynamical features of the decoherence functions?

Considered Hamiltonians in the ensemble are time independent and the features of the corresponding probability distribution allows then to quantify nonclassicality. What would happen to the definition of nonclassicality here if one allowed time-dependent Hamiltonians in the ensemble, i.e., one used fixed probability distribution in the initial point of time to draw from an ensemble of time-dependent Hamiltonians delivering each time evolution?

When the figures are cited in the text, often the corresponding numbers of figures are missing.

Reviewer #2 (Remarks to the Author):

The authors face the problem of quantifying the non-classicality (or quantumness) of the evolutions of a quantum system (pure dephasing, specifically). This is an interesting point, since it is known that pure dephasing dynamics of quantum systems can be simulated by classical Hamiltonian ensembles. The aim also lays at the heart of identifying the classical-quantum border. They find that it is possible to reach a quantitative measure for the classicality of pure-dephasing dynamics by introducing suitable (quasi)-probability distribution and a proper distance measure, which has an operational interpretation.

I find the results of a certain interest. However, as a general point, the authors already found the qualitative assessment for nonclassicality of system dynamics by HE with negative quasi-probability distributions (Ref. [14]). Here they extend this concept to be quantitative, giving a degree to the negative contributions, specifying to pure dephasing. The importance of Ref. [14] for the present work is remarked by the several recalls to it along the manuscript. Therefore, I suggest the authors to make an effort to improve the motivation of this work in order to clarify why this new paper is particularly innovative and of impact with respect to the previous one, such to guarantee publication in Nature Communications.

Besides this, I have some other technical comments, listed in the following.

1) Page 1, Abstract. When the authors write: "Dephasing processes, caused by the information exchange between [...]", I suggest to add the adjective "non-dissipative" before "information exchange". This slight modification would add clarity to the description, since system-environment information exchange can occur in a dissipative way causing decoherence (decay of system excitations) instead of dephasing alone. Such a change should be then made in the relevant points in the main text.

2) Page 2, Abstract, second last line. I would change the word "legitimate" with "classical" (or adding "classical" maintaining "legitimate") to highlight the contrast with the non-classical (quantum, i.e. negative) probability distributions (quantum dephasing processes).

3) Page 8, second line after Eq. (4). From the main text, it is not clear which is the meaning of n^{-1} in the definition of ρ . From the Supplementary Note 2, one can see that this is related to the coefficient of the identity matrix which is one of the operators defining ρ . However, for the sake of consistency of the main text, I suggest the authors to be explicit here in the definition of ρ .

4) Page 9, lines 1-4 from the top. Concerning the classicality of the reduced system dynamics when system and environment always remain classically correlated, the authors may suitably make reference to the well-known problem of the characterization of the system dynamics under classical environments in absence of back-action. In fact, the problem of system-environment correlations, classical environments and back-action is strictly linked to the aim of finding a classical or quantum measure of a system dynamics. To help the authors in retrieving useful literature, I may suggest the following works:

- Physical Review A 85, 032318 (2012).
- Nature Communications 4, 2851 (2013).
- Annals of Physics 350, 211 (2014).

5) Page 11, central paragraph (lines 10-14). There are some typos in this paragraph to be corrected. In particular, "spin-boson" should require "model" and there is a missing reference to the figure at the end of the paragraph.

6) Page 16, line 3 from the top. Once again, a correct reference to the corresponding figure is missing (it should be Fig. 3a). The same typos occur for other reference to the figures along the

paper.

7) Page 19, lines 1-3 from the top. I find the final discussion a bit too much technical to be suitable for the journal. In particular, I suggest the authors not to repeat the technical details which led them to find the main results. Instead, they could focus more on the fundamental importance of their introduced measure and the prospects. They just conclude mentioning the potential application in the theory of decoherence-free subspace, which they already discussed at the end of the previous section. I reckon that a discussion about the possible implications of the results to different contexts would be more useful here.

In conclusion, I believe that the manuscript can be considered for publication provided that the authors positively address the points above.

Reviewer #3 (Remarks to the Author):

Background.

Benchmarking quantum dynamics is a relevant task which faces numerous difficulties, both at the very fundamental level and also from the practical point of view. Provided a set of experimental data, the question is to assess whether a classical (according to a certain definition) theory can account for the observations or whether those necessarily rely on a quantum description. In the latter case, we expect that capabilities that were absent in a classical setting may now manifest. It is clear from this general considerations that the question of defining and quantifying "classicality" is likely to involve different aspects that may or may not be captured by a single measure. A similar situation is encountered when addressing the presence or absence of memory in the evolution. The current manuscript builds up on a recent paper by the same authors and published in Physical Review Letters in 2018. There exist also older results by others, most notably Audenaert and Steel NJP 2005 with a very similar approach. In both papers, the system's dynamics is defined to be classical if it can be represented as a convex sum of unitaries. The main result in PRL 2018 is providing a criterium for assessing the classicality of the system dynamics in terms of the existence of a genuine (ie positive) probability distribution for the unitaries reproducing the dynamics map describing the system dynamics. The fact that in some cases it is possible to draw a connection to the type of SE correlations that build up during the evolution provides an interesting physical insight to the approach.

Results.

In the current manuscript the formalism developed in PRL 2018 is extended to providing a quantitative measure of non-classicality for purely dephasing evolutions. The considered measure is formulated in terms of a variational distance, eq. (5). Group theoretical tools are employed to provide a constructive protocol for its computation which is then illustrated for qubits, qutrits and two non interacting qubits coupled to a common bath.

Comments.

By construction, the emphasis of this approach relies on the effect of the system-environment (SE) interaction on the reduced evolution, while the system dynamics, which may concern a composite system and have a non zero entangling power, is irrelevant. Therefore, the main question on classicality versus quantumness could be rephrased as assessing whether or not the system dynamics can be modelled by using a classically fluctuating field or whether that modellization is untenable, in which case the environment would be a source of quantum noise. These considerations can be viewed as merely semantic but I think it would be important to explain in the introduction that a related problem, which is actually closer in spirit to the current citation to Leggett inequalities, would be assessing the classicality of a given dynamical map that is enforcing a certain evolution and whose entangling or coherence power can be affected by the SE interaction. This is a related issue that has been discussed recently by Smirne et al in Quantum Sci. Technol. 4, 01LT01 (2019), Knee et al in Phys. Rev. A 98, 052328 (2018) and Milz et al, rXiv:1712.02589. The very related work by Audenaert and Scheel in New Journal of Physics 2008 is not cited in this paper. It is very likely that any potential reader will have to first read the previous work by the authors in PRL 2018 to actually understand the current manuscript and there

is a citation there but this is clearly insufficient, more when noting that these authors already introduced a distance measure to quantify non classicality. Personally, I would have welcome a much more detailed comparison between the formulation in PRL2018 and these results but at least some clarifying sentences must be included for fairness and to understand the connection between both approaches. It is entirely unclear to me how does the complexity of the evaluation of the classicality measure scales with the system dimension. Audenaert and Scheel provide a numerical algorithm. In the current case, the protocol needs to have access to the structure of the map so it seems necessary to actually perform full process tomography. Is this correct? Even if the procedure would be unrealistic for highly dimensional systems, an implementation with a single-few qubits system could be of interest. The novelty and scope of the manuscript would be largely enhanced if the authors would discuss a possible architecture where to implement the proposed method and analyze the resulting information. Superconducting qubits seems a feasible option given the background of some of the authors and there exist experimental results for process tomography in those systems. See for instance Phys. Rev. B 82, 184515 (2010). Can one benchmark the presence of quantum correlations between SE in this architecture? Is the noise mainly classical? Is indeed the proposed framework to actually learn something new in a real experiment?

Recommendation.

In my opinion, and in the light of the comments above, the manuscript does not contain yet enough new physics to warrant publication in a broad scope, very high profile journal. Should the work be supplemented by an actual experiment or provide at least a detailed analysis of a realistic model where the proposed measure is actually implemented and showed to provide significant information, my recommendation would be different.

>Reviewers' comments:

>

>Reviewer #1 (Remarks to the Author):

>

>Open system dynamics and decoherence studies are important fields of physics. Within this >framework, the manuscript demonstrates how to quantify nonclassicality of dephasing >processes in generic manner. Even though there exists a few previous studies on the >nonclassicality of quantum processes (cited in the manuscript), and some of the authors >themselves have recently studied nonclassicality of the dynamics especially in the association of >the type of the generated system-environment correlations (ref. 14), the results presented here >are highly non-trivial and display significant results for the studies of open quantum systems. >The nonclassicality measure introduced in Eq. (5) is conceptually rather simple. However, to >demonstrate that this measure is meaningful and useful in quantifying nonclassicality require >elaborate mathematical considerations and calculations, as the material of the manuscript >shows, reflecting the nontriviality of the results. Once this has been achieved a >general way to quantify nonclassicality of dephasing becomes available.

>

>In broader view, there has been during the recent years large amount of papers quantifying non->Markovianity of the open systems dynamics and elsewhere, e.g., a popular area of quantifying >quantum coherence has emerged. I think that the present results fit very well for this >background >and at the same time opens new directions in the open system studies in addition of >the importance of quantifying nonclassicality. Before making a definitive recommendation, I >would like the authors to address and comment the following points. I think this would improve >the readability of the paper considering the broad and various audiences it may have.

We thank reviewer #1 for his/her positive assessment of our manuscript.

>In ref. 14 some of the current authors mention that any nonunitary dynamics is classified as >nonclassical. Therefore I suppose that the current manuscript focuses on dephasing (unitary) >dynamics which may display classical and nonclassical features. Can the authors elaborate why >nonunitary dynamics should always be considered nonclassical and whether the current >construction, or a variant of it, could be useful or meaningful also for the nonunitary case?

This question raised by reviewer #1 is important, as it touches one of the core concepts of our theory. As stated in the main text, if the system and the environment remain classically correlated during the entire evolution, then the reduced system dynamics admits a time-independent Hamiltonian-ensemble description. This has been rigorously proved in our previous work (Ref. [17] in the revised manuscript) by using group theory. The present work is then based on the central implication that, whenever the reduced system dynamics cannot be simulated by using a Hamiltonian ensemble (i.e., one necessarily resorts to a quasi-distribution with negative values),

then the emergence of nonclassical correlations between the system and the environment is witnessed by the negative values in the quasi-distribution. The importance of this implication has been remarked in the manuscript that, to verify the emergence of nonclassical correlations, we merely need the knowledge of the system dynamics, which can be measured by using quantum process tomography, providing the feasibility of our approach.

On the other hand, as shown by Eq. (1), all ensemble-averaged dynamics are unital; therefore, nonunital dynamics do not admit such Hamiltonian-ensemble simulation. Consequently, nonunital dynamics are considered nonclassical according to our definition.

Additionally, the procedure presented in the current work relies on the underlying *abelian* algebraic structure (i.e., Cartan subalgebra) of pure dephasing; therefore the current formalism is not necessarily valid in general *non-abelian* cases of nonunital dynamics, wherein the Baker-Campbell-Hausdorff formulae are required.

It is worthwhile to note that, as a consequence, our methodology points out an intrinsic difference between pure dephasing and dissipative dynamics, in terms of the algebraic structures behind them. Such viewpoint may provide new insights into the theory of open system dynamics. Consequently, although the current formalism is limited to pure dephasing, our methodology based on the algebraic structure can be applied to the non-abelian case and therefore sheds light on a new avenue to study open system problems. This is also one of our following works in the future.

- >Suppose that one knows that the dynamics follows dephasing and has the subsequent
- >experimental data based on measurements available. In which way and by using what
- >resources one can then conclude whether this corresponds to classical or nonclassical
- >dynamics, and if the latter, then how much?
- >Expressed in slightly different way: the manuscript mentions that the nonclassicality measure is
- >developed here in the spirit of Wigner function and its negativity. Now one can reconstruct the
- >Wigner function based on experimental data. How does one reconstruct the probability
- >distribution for the purpose of nonclassicality based on the experimental data only?

Our approach can be implemented with experimental data measured by quantum process tomography or quantum state tomography experiments on the reduced system dynamics. In the revised manuscript, we now discuss how one can reconstruct the dynamical linear map [i.e., the matrix on the left hand side of Eq. (3)] from the χ matrix measured by quantum process tomography. Note that this reconstruction of the dynamical linear map is generally valid, not restricted to pure

dephasing nor unital dynamics. Assuming that the dynamics follows pure dephasing, one can proceed to retrieve the (quasi-)distribution function [on the right hand of Eq. (3)] such that Eq. (3) is satisfied, by following the procedure presented in the manuscript.

Furthermore, we have also demonstrated a numerical simulation of a quantum state tomography experiment on the S-T₀ qubit in double quantum dots, and also incorporating experimental errors. In such experiments, we first use the data of the return probabilities measured by projecting the quantum state onto the three axes of the Bloch sphere (quantum state tomography). Then we can depict the state trajectory in the Bloch sphere. This clearly visualizes the pure dephasing dynamics of the S-T₀ qubit and eases the burden of analyzing the huge data generated by the quantum process tomography experiment. Next, we can analyze the data step by step as explained in the Methods section and the newly added Supplementary Note 9 and obtain the final result.

These examples of manipulation with measured data underpin the practical feasibility of our theory.

>The authors justify their results by their generality compared to some earlier results, e.g., ref. 13
>deals with single bosonic mode. On the other hand, the considered examples in the current
>contribution are qubit, qutrit, and qubit-qubit systems, i.e., no continuous variable systems are
>used as examples here. Can the authors say something more about the applicability of their
>results, e.g. to harmonic oscillator as open system case?

This question in fact clarifies the merit of our contributions and what our examples are exactly meant to be. To our knowledge, the notion of nonclassicality has been firstly introduced for bosonic systems, namely continuous variables, such as squeezing, sub-poissonian statistics, etc. For infinite-dimensional systems, non-positive quasi probability distributions, e.g. Wigner function, Glauber-Sudarshan P function, are generally an indication of nonclassicality. For instance, coherent states are classical since their distributions over the phase space are Gaussian and positive, whereas entangled Gaussian states or squeezed states are non-classical since their P functions are not positive.

The notion of non-classicality defined for continuous variable systems is however limited in the sense that it quantifies that all finite-dimensional systems are non-classical. For instance, a single photon state is forever nonclassical. Although one can admit nonclassicality of finite-dimensional systems in general, an elaborated notion of nonclassicality for finite-dimensional systems is needed for useful quantifications. For finite-dimensional systems, entanglement of qubits, correlations

between qubits and a bath, or error-correctability of qubit states, etc. are of great significance, in particular for quantum information applications. Note that entanglement of finite dimensional systems has been extensively investigated since it is a resource for quantum information processing.

Our contribution here develops the nonclassicality of dynamical processes in the view of correlations between a system and a bath. The nonclassicality vanishes when a bath can have access to full information about a system, hence, an information-theoretic quantification. The notion of nonclassicality can be in principle applied to infinite-dimensional systems, e.g. a harmonic oscillator. The only non-triviality is the fact that there are infinite number of orthogonal states (number states) and one may need a compact representation such as coherent states, which are however not orthogonal, thus our quantification is not straightforwardly applied. We, however, point out that it is an interesting future investigation to consider how our notion of nonclassicality is related to quasi probability distributions for continuous variable systems. In the revision, we add a line in the Discussion section "we also find that it would be interesting to investigate how the notion of dynamical process nonclassicality is related to other quasi-distributions [63]".

>In example cases, the results demonstrate how the negativity of the used probability distribution
>shows up when changing certain parameters. However, I do not see results or plots on the
>decoherence functions and how they behave in different cases. Thereby the question is whether
>the classicality displays itself in the evolution of the coherences? Or whether nonclassicality of
>the dynamics can be associated to specific dynamical features of the decoherence functions?

The nonclassicality of the dynamics in our work originates from the establishment of nonclassical correlations between the system and the environment during their time evolution, and manifests itself in terms of the negativity of the quasi-distribution function coming with the simulating Hamiltonian ensemble. However, it is generically not easy to verify the properties of bipartite correlations from reduced system coherence alone, without accessing the environment. In other words, one cannot infer the nonclassicality of the dynamics by exclusively investigating the individual behavior of the reduced system coherence.

In order to study the nonclassicality of the dynamics, we must consider all the dephasing factors $\{\phi_m(t)\}_m$ collectively according to the underlying algebraic structure of pure dephasing, rather than focusing on their individual behavior. To this end, we establish a *formally generalized* inverse Fourier transform from the group of dephasing factors to the quasi-distribution, which allows us to decompose such an inverse-transform problem into a set of coupled equations defined along the root

vector axes. The resulting quasi-distribution function is the interplay between all the dephasing factors.

>Considered Hamiltonians in the ensemble are time independent and the features of the
>corresponding probability distribution allows then to quantify nonclassicality. What would happen
>to the definition of nonclassicality here if one allowed time-dependent Hamiltonians in the
>ensemble, i.e., one used fixed probability distribution in the initial point of time to draw from an
>ensemble of time-dependent Hamiltonians delivering each time evolution?

We agree with reviewer #1 that a Hamiltonian ensemble with explicit time dependence would be an interesting generalization. In particular, as discussed in our previous work [PRL **120**, 030403 (2018)], all unital *qubit* dynamics can be simulated with time-dependent Hamiltonian ensembles. This is a conclusion of Ref. [48] in the revised manuscript. However, we have rigorously proved that a persistently classical bipartite correlation definitely leads to a time-independent Hamiltonian ensemble decomposition for the reduced system dynamics. Based on this result, the strictly time-independent Hamiltonian ensemble simulation possesses a clear physical meaning as a criterion: Any violation of the strictly time-independent Hamiltonian ensemble simulation, such as negativity in the quasi-distribution or any time-dependence in the ensemble, witnesses the emergence of nonclassical correlations and therefore can be considered as a signature of the nonclassicality of the dynamics.

Additionally, we believe that such definition with a time-independent ensemble is more practical as the necessity to demonstrate the nonexistence of a time-independent HE is arguably considerably less involved than to demonstrate the nonexistence of a time-dependent HE, as the latter requires to exclude a significantly vaster range of possibilities. Nevertheless, we agree that the generalization to the time-dependent case appears interesting in its own and might be pursued in the future.

>When the figures are cited in the text, often the corresponding numbers of figures are missing.

We thank reviewer #1 for pointing out this issue. We have fixed this in the revised manuscript.

>Reviewer #2 (Remarks to the Author):

>

>The authors face the problem of quantifying the non-classicality (or quantumness) of the
>evolutions of a quantum system (pure dephasing, specifically). This is an interesting point, since
>it is known that pure dephasing dynamics of quantum systems can be simulated by classical
>Hamiltonian ensembles. The aim also lays at the heart of identifying the classical-quantum
>border. They find that it is possible to reach a quantitative measure for the classicality of pure-
>dephasing dynamics by introducing suitable (quasi)-probability distribution and a proper
>distance measure, which has an operational interpretation.

We first thank reviewer #2 for his/her positive comments on our manuscript.

>I find the results of a certain interest. However, as a general point, the authors already found the
>qualitative assessment for nonclassicality of system dynamics by HE with negative quasi-
>probability distributions (Ref. [14]). Here they extend this concept to be quantitative, giving a
>degree to the negative contributions, specifying to pure dephasing. The importance of Ref. [14]
>for the present work is remarked by the several recalls to it along the manuscript. Therefore, I
>suggest the authors to make an effort to improve the motivation of this work in order to clarify
>why this new paper is particularly innovative and of impact with respect to the previous one,
>such to guarantee publication in Nature Communications.

We thank reviewer #2 for his/her constructive suggestion. We agree that the present work is conducted based on the previous work (Ref. [17] in the revised manuscript). However, the present work has provided a substantial breakthrough and unambiguously addressed several non-trivial issues, which cannot be tackled within the framework of the previous work. Therefore, we believe that the present work significantly promotes the practicality of our theory, going beyond the scope of Ref. [17].

Specifically, the connection between the nonclassicality of the dynamics and the Hamiltonian-ensemble simulation has been established in Ref. [17], and two examples of qubit pure dephasing were worked out for demonstrating such connection. However, as discussed in the conclusion section of Ref. [17], to prove the nonexistence of a Hamiltonian-ensemble simulation, and therefore to verify the nonclassicality of a dynamics, is generically non-trivial. Additionally, as discussed in Ref. [48], the closely related problem of random-unitary decomposition can merely be numerically implemented without an efficient analytic solution.

To provide an answer to this problem, in the present work, we derive a canonical form of the Fourier transform on groups, which is proven to be a powerful tool as it allows us to establish a constructive procedure and definitely prove its uniqueness for pure dephasing of any dimension by using group-theoretical tools. Based on these results, we can lift the quasi-distribution to be a faithful representation of pure dephasing and propose a quantitative measure of nonclassicality. These achievements are accomplished with the help of the Fourier transform on groups and cannot be realized within the framework of Ref. [17].

To highlight these significant advances in the present work, in the revised Introduction (and a few sentences in the revised Abstract), we have pointed out the technical difficulties of proving the nonexistence of simulating Hamiltonian ensembles. The present work has addressed this problem by establishing a constructive procedure and proving its uniqueness for the case of pure dephasing. Additionally, we are able to establish a quantitative measure of dynamics nonclassicality, going beyond the merely qualitative discrimination in our previous work.

Finally, to underpin the practical feasibility of our approach, we have shown the connection between the dynamical linear map and the experimental raw data, which can be measured by using quantum process tomography. Furthermore, we have numerically simulated a quantum state tomography experiment on the S-T₀ qubit pure dephasing in double quantum dots, also incorporating experimental errors. With this simulation, we explain how to implement step-by-step our approach on experimental raw data.

>Besides this, I have some other technical comments, listed in the following.

>1) Page 1, Abstract. When the authors write: "Dephasing processes, caused by the information >exchange between [...]", I suggest to add the adjective "non-dissipative" before "information >exchange". This slight modification would add clarity to the description, since system->environment information exchange can occur in a dissipative way causing decoherence (decay >of system excitations) instead of dephasing alone. Such a change should be then made in the >relevant points in the main text.

>2) Page 2, Abstract, second last line. I would change the word "legitimate" with "classical" (or >adding "classical" maintaining "legitimate") to highlight the contrast with the non-classical >(quantum, i.e. negative) probability distributions (quantum dephasing processes).

We thank reviewer #2 for his/her suggestions on the text. We have revised the Abstract according to the above two points.

>3) Page 8, second line after Eq. (4). From the main text, it is not clear which is the meaning of n^{-1} in the definition of ρ . From the Supplementary Note 2, one can see that this is related to the coefficient of the identity matrix which is one of the operators defining ρ . However, for the sake of consistency of the main text, I suggest the authors to be explicit here in the definition of ρ .

We have fixed this issue by explicitly expressing ρ in terms of a linear combination of the identity and Hermitian generators, wherein the coefficient n^{-1} is manifest. We hope that this modification improves the readability of the revised manuscript.

>4) Page 9, lines 1-4 from the top. Concerning the classicality of the reduced system dynamics when system and environment always remain classically correlated, the authors may suitably make reference to the well-known problem of the characterization of the system dynamics under classical environments in absence of back-action. In fact, the problem of system-environment correlations, classical environments and back-action is strictly linked to the aim of finding a classical or quantum measure of a system dynamics. To help the authors in retrieving useful literature, I may suggest the following works:

- >- Physical Review A 85, 032318 (2012).
- >- Nature Communications 4, 2851 (2013).
- >- Annals of Physics 350, 211 (2014).

We agree that these references are interesting and relevant. We have added them into the list of references in the revised manuscript. We believe that these references will further underpin the wider scope of our work.

>5) Page 11, central paragraph (lines 10-14). There are some typos in this paragraph to be corrected. In particular, "spin-boson" should require "model" and there is a missing reference to the figure at the end of the paragraph.

>6) Page 16, line 3 from the top. Once again, a correct reference to the corresponding figure is missing (it should be Fig. 3a). The same typos occur for other reference to the figures along the

>paper.

We thank reviewer #2 for pointing out these two issues. We have fixed them in the revised manuscript according to the above two points.

>7) Page 19, lines 1-3 from the top. I find the final discussion a bit too much technical to be
>suitable for the journal. In particular, I suggest the authors not to repeat the technical details
>which led them to find the main results. Instead, they could focus more on the fundamental
>importance of their introduced measure and the prospects. They just conclude mentioning the
>potential application in the theory of decoherence-free subspace, which they already discussed
>at the end of the previous section. I reckon that a discussion about the possible implications of
>the results to different contexts would be more useful here.

We agree with reviewer #2 that a suitable emphasis on the importance and prospect may further clarify the contributions achieved in the present work. In the revised Discussion section, we now succinctly summarize the important achievements of the present work. These include the establishment of a constructive procedure to retrieve the (quasi-)distributions for pure dephasing of any dimension, proving the existence and uniqueness provided commuting member Hamiltonians, the CHER as a faithful representation, and the quantitative measure of nonclassicality going beyond the merely qualitative discrimination. Moreover, we have also emphasized the viability of our approach by outlining its implementation based on quantum process tomography and quantum state tomography.

Additionally, as a prospect for following works in the future, we also discussed the generalization beyond pure dephasing and the difficulties one might encounter. Since our approach exploits the abelian algebraic structure of pure dephasing, generalizations going beyond pure dephasing will be difficult because of the non-abelian algebraic structure, which renders it inevitable to encounter the complicated Baker-Campbell-Hausdorff formulae. This, on the one hand, implies that our approach points out an essential discrimination between different types of dynamics in terms of their underlying algebraic structures. On the other hand, such group-theoretical analysis provides a powerful tool and may open a new route toward the investigation of open quantum system theory.

>In conclusion, I believe that the manuscript can be considered for publication provided that the
>authors positively address the points above.

We thank reviewer #2 for his/her positive comments and recommendation of our manuscript.

>Reviewer #3 (Remarks to the Author):

>

>Background.

>Benchmarking quantum dynamics is a relevant task which faces numerous difficulties, both at
>the very fundamental level and also from the practical point of view. Provided a set of
>experimental data, the question is to assess whether a classical (according to a certain
>definition) theory can account for the observations or whether those necessarily rely on a
>quantum description. In the latter case, we expect that capabilities that were absent in a
>classical setting may now manifest. It is clear from this general considerations that the question
>of defining and quantifying "classicality" is likely to involve different aspects that may or may not
>be captured by a single measure. A similar situation is encountered when addressing the
>presence or absence of memory in the evolution. The current manuscript builds up on a recent
>paper by the same authors and published in Physical Review Letters in 2018. There exist also
>older results by others, most notably Audenaert and Steel NJP 2005 with a very similar
>approach. In both papers, the system's dynamics is defined to be classical if it can be
>represented as a convex sum of unitaries. The main result in PRL 2018 is providing a criterium
>for assessing the classicality of the system dynamics in terms of the existence of a genuine (ie
>positive) probability distribution for the unitaries reproducing the dynamics map describing the
>system dynamics. The fact that in some cases it is possible to draw a connection to the type of
>SE correlations that build up during the evolution provides an interesting physical insight to the
>approach.

.

>Results.

>In the current manuscript the formalism developed in PRL 2018 is extended to providing a
>quantitative measure of non-classicality for purely dephasing evolutions. The considered
>measure is formulated in terms of a variational distance, eq. (5). Group theoretical tools are
>employed to provide a constructive protocol for its computation which is then illustrated for
>qubits, qutrits and two non interacting qubits coupled to a common bath.

We first thank reviewer #3 for his/her careful reading and the elaborate report.

>Comments.

>By construction, the emphasis of this approach relies on the effect of the system-environment

>(SE) interaction on the reduced evolution, while the system dynamics, which may concern a
>composite system and have a non zero entangling power, is irrelevant. Therefore, the main
>question on classicality versus quantumness could be rephrased as assessing whether or not
>the system dynamics can be modelled by using a classically fluctuating field or whether that
>modellization is untenable, in which case the environment would be a source of quantum noise.
>These considerations can be viewed as merely semantic but I think it would be important to
>explain in the introduction that a related problem, which is actually closer in spirit to the current
>citation to Leggett inequalities, would be assessing the classicality of a given dynamical map
>that is enforcing a certain evolution and whose entangling or coherence power can be affected
>by the SE interaction. This is a related issue that has been discussed recently by Smirne et al in
>Quantum Sci. Technol. 4, 01LT01 (2019), Knee et al in Phys. Rev. A 98, 052328 (2018) and
>Milz et al, rXiv:1712.02589.

We thank reviewer #3 for pointing out an alternative definition of nonclassicality from the viewpoint of entangling or coherence power and related works. This broadens the scope of the revised Introduction. We have added these references into the list of references.

>The very related work by Audenaert and Scheel in New Journal of Physics 2008 is not cited in
>this paper. It is very likely that any potential reader will have to first read the previous work by
>the authors in PRL 2018 to actually understand the current manuscript and there is a citation
>there but this is clearly insufficient, more when noting that these authors already introduced a
>distance measure to quantify non classicality.

We thank reviewer #3 for pointing out the missing reference, which discusses closely related problems of random-unitary decomposition and therefore supports the fundamental importance of our work. We have also added it into the list of references.

>Personally, I would have welcome a much more detailed comparison between the formulation in
>PRL2018 and these results but at least some clarifying sentences must be included for fairness
>and to understand the connection between both approaches.

The conclusions of our previous work are all formulated in terms of the mixture of unitary evolutions [cf. Eq. (1) in the present manuscript]. However, its practical utility in more general cases is limited since such formulation still relies on the input of an initial state. To completely shift the focus to the dynamics itself, we recast the formulation into the form of a Fourier transform on groups Eq. (3). Although Eqs. (1) and (3) are mathematically equivalent according to the Lie algebra adjoint representation, the Fourier transform formulation Eq. (3) derived in the present manuscript is a powerful tool. Utilizing these group-theoretical techniques, all the achievements in the manuscript stem from Eq. (3). These can hardly be accomplished within the framework of our previous work.

To clarify this point, in the revised manuscript, we have emphasized the advantage of Eq. (3) by elaborating that it provides further physical insights into the CHER and nonclassicality, and manifests its usefulness in our constructive procedure and the proof of uniqueness.

>It is entirely unclear to me how does the complexity of the evaluation of the classicality measure
>scales with the system dimension.

The dependence of the complexity on the system dimension can be understood from the underlying algebra. The quasi-distribution for a pure dephasing is obtained by solving the set of simultaneous Eqs. (8) in the revised manuscript, wherein the index m runs over all positive root vectors. For an n -dimensional system, both the density matrix and the member Hamiltonian operators are n -by- n matrices. Therefore the member Hamiltonian operators belong to the Cartan subalgebra of $\mathfrak{su}(n)$ Lie algebra, which possesses $n(n-1)/2$ positive root vectors. Therefore, the number of involved equations in Eqs. (8) grows with the square of the system dimension.

>Audenaert and Scheel provide a numerical algorithm. In the current case, the protocol needs to
>have access to the structure of the map so it seems necessary to actually perform full process
>tomography. Is this correct? Even if the procedure would be unrealistic for highly dimensional
>systems, an implementation with a single-few qubits system could be of interest. The novelty
>and scope of the manuscript would be largely enhanced if the authors would discuss a possible
>architecture where to implement the proposed method and analyze the resulting information.

We thank reviewer #3 for his/her suggestion of connecting to realistic architectures. As stated by the reviewer #3, the dynamical linear map characterizes the complete time evolution of the reduced system and can be reconstructed from the raw data measured by the quantum process tomography technique.

In the revised manuscript, we have explained how to recover the dynamical linear map from the raw data of $\chi_{Jm}(t)$ measured by the quantum process tomography, which has been reported on several platforms, including optical setups, trapped ions, and superconductors. Then the reconstructed map can be postprocessed according to our theory. Furthermore, it should be noted that, to reproduce the complete time evolution of the system, the raw data of $\chi_{Jm}(t)$ in a full time sequence is necessary. This is in principle attainable by repeating the experiment for a full time sequence without imposing additional technical difficulty.

On the other hand, as explained above, to perform quantum process tomography in a full time sequence requires extensive computational resources to handle the huge amount of raw data. Therefore, to further enhance the impact of our manuscript, we have also demonstrated a numerical simulation of the quantum state tomography experiment, which is relatively less demanding. In particular, if the reduced-system-dynamics follows pure dephasing, the two techniques provide equivalent information. In the revised manuscript, we have demonstrated the simulation of the S-T₀ qubit pure dephasing and numerically analyzed the return probabilities measured by quantum state tomography. Additionally, we have also taken experimental noise into account. We have shown the implementation of our approach with the measurable data step by step. We believe that this simulation exhibits the in-principle feasibility of our approach and is capable of stimulating follow-up experimental works.

- >Superconducting qubits seems a feasible option given the background of some of the authors
- >and there exist experimental results for process tomography in those systems. See for instance
- >Phys. Rev. B 82, 184515 (2010). Can one benchmark the presence of quantum correlations
- >between SE in this architecture? Is the noise mainly classical? Is indeed the proposed
- >framework to actually learn something new in a real experiment?
- >
- >Recommendation.
- >In my opinion, and in the light of the comments above, the manuscript does not contain yet
- >enough new physics to warrant publication in a broad scope, very high profile journal. Should
- >the work be supplemented by an actual experiment or provide at least a detailed analysis of a
- >realistic model where the proposed measure is actually implemented and showed to provide
- >significant information, my recommendation would be different.

In the revised manuscript, we have demonstrated a numerical simulation of the quantum state tomography experiment on the S- T_0 qubit pure dephasing in double quantum dots. In such experiments, the measured raw data are the return probabilities projected onto the three axes of the Bloch sphere. In our simulation, we show the qubit pure dephasing at two different exchange (J) values, which can be adjusted by varying the detuning between the two quantum dots. In the following, we explain step-by-step the manipulation of the measured raw data.

Similar to the situation in quantum process tomography, the measurements should be performed repeatedly for different values of the free induction decay time τ_s , to generate the raw data of return probabilities as time sequences, as shown in the Fig. 4a in the revised manuscript.

Based on the measured raw data of return probabilities, we can recover the time evolution of the density matrix according to the equations explained in the revised Method section. Meanwhile, we can also depict the trajectories in the Bloch sphere according to Eq. (19), as shown in Fig. 4b. This clearly visualizes the dynamics, and helps us to identify the axis of rotation.

According to the axis of rotation identified in the Bloch sphere, we can easily recover the standard form of pure dephasing in Eq. (2) by choosing a suitable basis transformation. Then our procedure is applicable and leads to Eq. (20). The final numerical solutions are shown in Fig. 4c. Several points can be discovered from Fig. 4c. First, the $\wp(\omega)$ mainly distributes over the region of negative ω due to the fact that the eigenenergy of the singlet $|S\rangle$ state is lower than the triplet $|T_0\rangle$ state; i.e., $\omega_S - \omega_{T_0} < 0$. Second, the shorter the coherence time, the broader the $\wp(\omega)$. This point is also in line with physical intuition.

Additionally, in order to further bridge the gap between our theoretical analysis and experiments, we have also simulated the experimental errors by dressing the theoretical model with Gaussian noise. This allows us to confirm the robustness of the nonclassicality detection against experimental errors in a realistic assessment.

Quantum state tomography experiments are usually less involved than the ones involving quantum process tomography. In particular, in the case of qubit pure dephasing, they provide equivalent information, with the help of Bloch sphere visualization. Consequently, with this demonstration, we believe that we have achieved to showcase the implementation of our approach on a realistic model, and prove the in-principle practical feasibility of our approach, which can provide new physical insights to the community of open quantum systems.

We hope that we have addressed all points raised by the reviewers and we believe that the revised manuscript is eligible for publication in Nature Communications. Finally, we conclude the revision with the following list of changes:

1. In the Introduction, we discuss the significant difficulty encountered in our previous work and summarize our current major achievements, which cannot be addressed within the framework of our previous work.
2. We compare and elaborate the advantage of the newly derived Fourier transform formulation over previous work.
3. We explain the implementation of our approach with the χ matrix measured by quantum process tomography. In addition, we demonstrate a numerical simulation of the quantum state tomography experiment of the S-T₀ qubit incorporating experimental errors, with which our approach is performed step by step.
4. In the Discussion, we again emphasize the significance of our conclusion and discuss several possible directions of our future works, including the generalizations going beyond pure dephasing, the algebraic analysis in open system theory, and the connection to other notions of nonclassicality.
5. We revise the Abstract according the reviewer #2 and the corresponding changes also added to the main text.
6. We add and discuss all the missing references suggested by the reviewers.
7. We slightly modify the presentation of the formulation according to reviewer #2.
8. We fix all the known typos and missing figure references in the main text.

Reviewers' comments:

Reviewer #1 (Remarks to the Author):

I have read in detail the response by the authors and the modified manuscript including the implemented changes to the text. Based on this and the points mentioned in the earlier report, I recommend the publication of this paper in Nature Communications.

Reviewer #2 (Remarks to the Author):

I have reviewed the revised version of the manuscript in object. To my opinion, the manuscript has been improved from both presentation and technical viewpoints. My previous comments and criticisms appear to be fulfilled. I believe the manuscript can be accepted for publication.

Reviewer #3 (Remarks to the Author):

In my view, the authors have addressed those comments in a satisfactory manner and despite the scope of the paper would have been significantly enhanced with an actual experiment, I am inclined to recommend publication. In my view, the topic is timely, the considered algebraic methods have the potential to be useful in other problems and I agree with the authors that the work is likely to motivate experimental work.